# The Impact of Digital Marketing Innovation on Firm Performance: Mediation by Marketing Capability and Moderation by Firm Size

**Sang-Uk Jung * and Valeriia Shegai**

Business School, Hankuk University of Foreign Studies, Seoul 02450, Republic of Korea
* Correspondence: sanguk.jung@hufs.ac.kr

**Abstract:** Digital marketing innovation plays an important role in a company's performance. Since this concept is quite new, there are not many empirical studies on the impact of marketing innovations. The purpose of this study is to examine the impact of digital marketing innovation on firm performance, consider the mediation effect of marketing capability on the impact of digital marketing capability on firm performance and explore the potential moderating effect of firm size on the mediation effect. Using KOSPI and KOSDAQ data and a linear moderated mediation estimation, we found that digital marketing innovation on firm performance through marketing capability has significant direct and indirect effects, with indirect effects greater than direct effects. Theoretical and practical implications are also discussed in this article.

**Keywords:** digital marketing innovation; firm performance; marketing capability; firm size

## 1. Introduction

Digital marketing innovation has become increasingly popular and important in recent years due to the growing reliance on digital channels for business and consumer communication. With the widespread adoption of digital technologies and the increasing amount of time people spend online, companies must adapt to this shift in consumer behavior by embracing digital marketing innovations. These innovations offer companies new opportunities to reach and engage with their target audiences, gather valuable data, and stay ahead of the competition. As a result, many companies are investing in digital marketing innovation and incorporating it into their overall marketing strategies to remain relevant and competitive in the digital marketplace [1,2].

Digital marketing innovation has the potential to positively impact firm performance by allowing companies to reach wider audiences, enhance customer engagement, and gather valuable data for targeted advertising. By embracing new technologies, companies can increase brand awareness, drive sales, and improve customer experience. Effective digital marketing innovation can lead to improved customer acquisition, retention, and loyalty, resulting in increased sales and overall firm performance [3,4]. Additionally, most researchers agree that digital innovation can be a powerful and an effective tool for promoting and creating sustainability [5]. The implementation of modern digital marketing tools enables companies to ensure sustainable development (improved customer service, customer focus, etc.) and thus achieve high financial performance [6]. Companies that prioritize sustainability and invest in digital innovation tend to outperform their peers in terms of financial performance. For example, a study by MIT Sloan Management Review found that companies that prioritize sustainability and digital innovation had a 16% higher profit margin than their peers.

However, the relationship between digital marketing innovation and firm performance is complex and interrelated with other factors such as firm size and marketing capability [7,8]. Firm size can affect a company's ability to invest in and implement digital marketing innovations [9,10]. Smaller firms may have limited resources, while larger firms

may have more resources but face challenges in adapting to new technologies [11,12]. Marketing capability, in terms of human and financial resources, technology, and strategy, can impact a firm's ability to successfully implement digital marketing innovations and drive performance [13,14]. Ultimately, the effectiveness of digital marketing innovation in improving firm performance depends on a variety of factors, including the company's marketing capability, firm size, and the specific digital marketing innovations being implemented, which have not yet been studied. Based on the above research gaps, this study answers the following research questions: (1) Does marketing capability mediate the impact of digital marketing innovation on firm performance? (2) Does firm size affect the mediating effect of marketing capability on the impact of digital marketing innovations on firm performance?

To answer the above research questions, we use the data of the top 100 companies of the Korean Composite Stock Price Indexes (KOSPI) and the Korean Securities Dealers Automated Quotations (KOSDAQ). Overall, while the United States is certainly a major player in digital innovation, there has been a growing interest in studying digital innovation in other countries, including South Korea, China, Japan, and European countries. This reflects the growing recognition that digital innovation is a global phenomenon that requires a global perspective. For years, Korea has been at the forefront of digital innovation, and its digital innovation ecosystem has served as a model for other countries seeking to develop their own digital industries, making it an excellent case for digital innovation research. To identify the complicated relationship between digital marketing and firm performance, this study considers the indirect effect of marketing capability on firm performance, that is, the mediation effect of marketing capability on the impact of digital marketing capability on firm performance. In addition, we explore the potential moderating effect of firm size on the mediation effect.

We organize this article as follows: First, we briefly review the literature related to digital marketing innovation, firm performance, marketing capability, and firm size and then develop hypotheses about how these constructs are related. Second, we describe the data and materials we used. Third, we present the results and discuss the implications of our findings. Finally, we explore the limitations of the current study and directions for further research.

## 2. Literature Review and Hypotheses Development

### 2.1. Digital Marketing Innovation and Firm Performance

In today's highly competitive world, markets have become increasingly global and technologically sophisticated while customers have become more demanding [15]. This has made it more challenging for businesses to compete and succeed in their respective industries. With the rise of technology, businesses can now easily connect with customers around the world, which has led to increased competition [16]. Customers also have access to more information than ever before, making them more informed and discerning about the products and services they purchase [17]. To succeed in this environment, businesses need to adapt to these changes and become more innovative in their approaches to marketing and customer service [18].

Today there are two types of marketing: traditional and digital marketing [6,19]. Traditional marketing refers to the more traditional methods of marketing that have been used for decades, such as television commercials, print ads, billboards, and radio spots. These methods typically involve reaching out to a wide audience in a specific geographic area or demographic group [20]. Digital marketing, on the other hand, refers to marketing methods that rely on digital technologies, such as social media, email marketing, search engine optimization (SEO), and paid online advertising. These methods are often more targeted and can reach specific audiences based on their interests, behaviors, or geographic locations [20,21]. While traditional marketing methods can still be effective, digital marketing has become increasingly popular and important in today's world due to the rise of technology and the Internet [22]. Digital marketing allows businesses to reach a wider

audience, measure their marketing efforts, and adjust their strategies based on data-driven insights [23]. It can be a cost-effective way to market products and services, and it offers the potential for greater ROI compared to traditional marketing methods [24,25]. In addition, social media is nowadays a key element of companies' marketing strategies. There is a strong correlation between customers' use of social media and their brand loyalty; a high level of usage yields higher loyalty. Moreover, a higher level of loyalty leads to better sales performance [26]. However, Pollák and Markovič conclude in their research that most organizations are not ready to invest in marketing communications based on digital marketing, because it is still perceived by many firms as a supplement to traditional marketing [27].

Innovation is defined as the process of introducing new ideas, products, or methods to improve an existing system or to create a new one [28]. It involves using creativity and problem-solving skills to develop novel solutions that can meet the changing needs of society [29]. There are different types of innovation. In our study, we will look at marketing innovation. Marketing innovation is the introduction of a new method of promoting, evaluating, or selling goods/services, or making significant changes in the aesthetic design or packaging of goods [30]. Digital marketing is one of the most important drivers of innovation leading to business competitiveness and growth [31,32]. As more and more consumers turn to digital channels to research and purchase products and services, businesses must embrace digital marketing to stay relevant and competitive in today's market. In Korea, in a highly competitive environment, consumers have a wide range of alternatives to meet their needs [33]. To be successful, Korean firms now require strong marketing management systems, including new product launches, effective promotions, and increased loyalty to attract customers and gain customer satisfaction to generate profits [33].

Prior research has repeatedly shown that innovation has a positive impact on a variety of firm performance by increasing customer satisfaction [34], production speed [35], and growth and efficiency in the example of a study of 1000 Fortune companies [36], along with increased sales, internal efficiency, and reduced production costs noted in other studies [37,38]. Empirical research on marketing innovation has also consistently shown that it has a positive impact on firm performance related to sales growth, profit, cash flow, and shareholder value [39–41]. Chung et al. in their work examining the top 100 Korean firms proved that by putting great effort in managing social media, a firm achieves high financial performance [42]. Marketing innovations positively affect firm performance because they can grant firm a more profitable competitive position in the market [43], and the economic efficiency brought about by marketing innovation constitutes a sustainable competitive advantage, contributing to product differentiation and increased consumption [44]. Based on previous studies, this study proposes the following hypothesis:

**Hypothesis 1.** *Digital marketing innovation has a positive impact on firm performance.*

*2.2. The Mediating Role of Marketing Capability*

Marketing capability is an organization's ability to develop and implement effective marketing strategies and tactics to promote its products or services [45], build brand awareness [46], and generate revenue [47]. It involves a range of skills, knowledge, and resources related to market research, product development, branding, advertising, sales, and customer engagement [15]. Effective marketing capability is critical for businesses of all sizes and in all industries, as it allows them to reach and engage with their target audiences, differentiate themselves from competitors [48], and drive sales and revenue growth [47]. Strong marketing capability can also help a company adapt to changing market and customer needs [49], stay ahead of trends, and innovate new products or services.

The resource-based view (RBV) is a strategic management framework that suggests that a firm's unique combination of internal resources and capabilities is a key determinant of its sustained competitive advantage and long-term performance [50]. The RBV approach

suggests that a firm's competitive advantage depends on how well it can manage and leverage its resources and capabilities [51]. Capability is the ability to use a resource or organizational process created and developed within a company [52]. To develop the marketing capability of a company, it is necessary to properly combine the individual skills of employees and the available resources of the company [53]. The mere possession of digital resources is not sufficient to create value and competitive advantage, so the focus should be on how to use digital resources as inputs, which in turn create capabilities [54]. A firm that spends more resources on interacting with customers can improve its ability of "market sensing" [55]. As a valuable, rare, inimitable and nonsubstitutable resource [56], innovation enables enterprises to develop and maintain competitive advantages [57] as well as develop marketing capability critical to a firm's competitiveness [58].

The relationship between digital marketing innovation and marketing capability is complex and can depend on a variety of factors, including the specific context of the organization and the nature of the innovation being introduced. Digital marketing innovation can enhance marketing capability by enabling organizations to reach and engage with customers through new and innovative channels more effectively. For example, social media marketing and search engine optimization can help organizations expand their reach and connect with customers in more targeted and personalized ways [59,60]. Similarly, the use of data analytics and artificial intelligence can help organizations better understand customer behavior and preferences, enabling them to tailor their marketing strategies to better meet customer needs [61,62].

However, introducing digital marketing innovations can also create challenges for organizations, particularly if they lack the necessary resources, skills, or infrastructure to effectively implement and manage these innovations. For example, implementing new digital marketing tools and platforms can require significant investments in technology and personnel as well as changes to organizational processes and structures. In some cases, organizations may also face resistance to change from employees or customers who are not familiar with the new technologies.

Prior research has repeatedly demonstrated that marketing capability has a positive impact on firm performance [33]. The results of an empirical study by Joensuu-Salo et al. prove that marketing capabilities have a direct impact on firm performance [63]. Marketing capability and firm performance are closely related, as a company's ability to effectively market its products or services is a critical factor in its overall success [64]. Effective marketing capability can lead to increased revenue, customer engagement, and market share [65], which can ultimately result in improved firm performance.

Marketing capability can affect firm performance in several ways. Firstly, strong marketing capability can help a company differentiate itself from its competitors and build a strong brand identity, which can increase customer loyalty and willingness to pay premium prices for the company's products or services [66–68]. Secondly, effective marketing capability can help a company identify and capitalize on new market opportunities and develop and launch new products or services that meet the needs and preferences of its target customers [69]. This can lead to increased revenue and market share, and ultimately improve the firm's financial performance [47]. Thirdly, marketing capability can help a company build strong relationships with its customers and stakeholders through effective communication, engagement, and customer service [53]. This can increase customer satisfaction and loyalty, reduce customer churn, and lead to positive word-of-mouth marketing, which can ultimately result in improved firm performance [70,71].

Therefore, we put forward the following hypotheses that marketing innovations have a positive impact on marketing capabilities:

**Hypothesis 2.** *Digital marketing innovation has a positive impact on marketing capability.*

**Hypothesis 3.** *Marketing capability has a positive impact on its firm performance.*

Marketing capability can indeed mediate the relationship between innovation and firm performance. Innovation is a key driver of firm performance [72], as it enables companies to develop new products or services, improve existing ones, and capture new market opportunities [73]. However, innovation alone may not necessarily lead to improved firm performance, as it needs to be effectively marketed and communicated to customers to generate revenue and achieve a competitive advantage [74–76].

Marketing capability plays a critical role in bridging the gap between innovation and firm performance. A company with strong marketing capabilities can effectively communicate the value and benefits of its innovative products or services to its target customers [49], generate demand and interest, and ultimately drive revenue growth and improved financial performance [47].

In addition, marketing capability can help a company identify and address customer needs and preferences [77,78] and tailor its marketing strategies to different customer segments [79]. This can lead to increased customer satisfaction and loyalty, reduced customer churn, and positive word-of-mouth marketing, all of which can contribute to improved firm performance [53,80].

Overall, marketing capability is a key mediator of the relationship between innovation and firm performance, as it enables companies to effectively market and communicate the value of their innovations to customers, generate demand and revenue [41], and build strong customer relationships that contribute to sustained competitive advantage and long-term success [50]. Hence, we propose the following hypothesis:

**Hypothesis 4.** *Marketing capability mediates the relationship between digital marketing innovation and firm performance.*

### 2.3. Moderating Effect of Firm Size

Prior literature suggests that the impact of innovation on a firm's business performance varies according to characteristics of the firm such as the firm's age, the firm's industry, company size, financial capital, entrepreneurial orientation, etc. [68,81].

Research has shown that innovation can be a powerful tool for driving growth and improving performance for smaller firms [10,82,83]. Smaller firms often have more limited resources and face more competition, and thus innovation can help them to differentiate themselves and gain competitive advantages [10]. Smaller firms are more nimble and able to respond quickly to changes in the market, making them better positioned to take advantage of new opportunities created by innovation [84–87]. Additionally, smaller firms tend to be more innovative than larger firms [88] due to a number of factors such as a more entrepreneurial culture, greater flexibility, and a higher tolerance for risk [89,90]. This allows smaller firms to implement new ideas more quickly, which leads to faster growth and improved performance with fewer resources [9,91].

However, for larger firms, the relationship between innovation and performance is more complex [9]. While larger firms have greater resources to invest in innovation, they face more internal resistance to change and have more complex decision-making processes [92]. In addition, larger firms have established routines and structures that can make it harder to implement new ideas and innovations [93].

Therefore, it is important to consider the role of firm size when exploring the relationship between innovation and firm performance. Hence, we propose Hypothesis 5:

**Hypothesis 5.** *The direct and indirect effects of digital marketing innovation on firm performance through marketing capability have a more positive impact on small and medium-sized firms than on large firms.*

The five research hypotheses formulated above have been integrated into a conceptual model of our study of how digital marketing innovation has an impact on firm performance through marketing capability moderated by firm size, as shown in Figure 1 below.

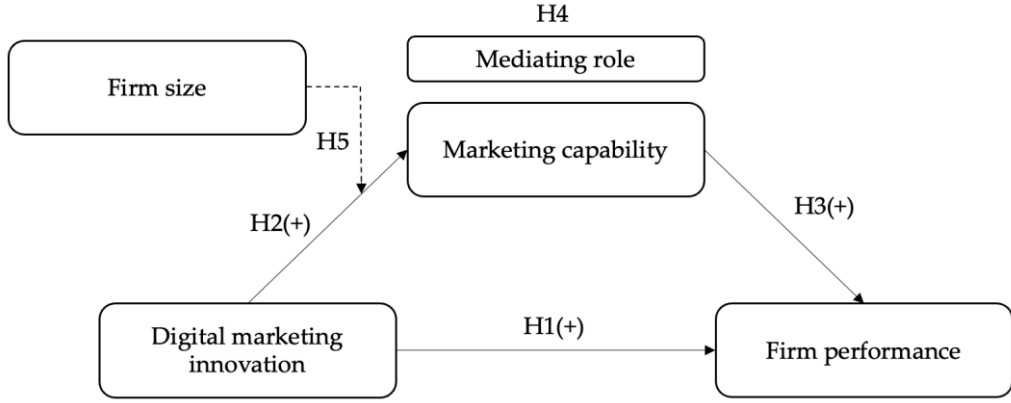

**Figure 1.** Conceptual Framework.

## 3. Materials and Methods

### 3.1. Sample and Data Collection

We tested the proposed model using data from the top 200 companies in KOSPI (Korea Composite Stock Price Index) and KOSDAQ (Korean Securities Dealers Automated Quotations) as of 29 December 2022. KOSPI is the main stock market index in South Korea. It represents the performance of the 200 largest and most liquid companies listed on the Korea Exchange (KRX) based on market capitalization [94]. KOSDAQ, on the other hand, is a stock market index that tracks the performance of smaller and emerging companies with high-growth potential, including startups listed on the KRX. Because KOSPI primarily consists of large-cap companies and KOSDAQ is focused on small and mid-cap companies with high-growth potential, this study used KOSPI and KOSDAQ as dummies to classify large firms and SMEs.

Data were collected from the KRX data information system (https://data.krx.co.kr, accessed on 1 November 2022), which provides useful statistical information in various fields from listed stocks to indices, securities products, derivatives, and general products.

Moderated mediation analysis [95] was conducted to examine the impact of digital marketing innovation on firm performance through the indirect effect of marketing capability, which was presumed to be moderated by firm size.

### 3.2. Measures

Firm Performance (*FP*): To measure firm performance (*FP*), the focal variable of this study, we selected Tobin's Q because Tobin's Q is widely used as a proxy for firm performance by practitioners and researchers [96,97]. In addition to Tobin's Q, there are other measures that can be used to assess firm performance such as revenue, profit, return on investment (ROI), and earnings per share (EPS). In general, a higher Tobin's Q ratio indicates better firm performance, as it suggests that the market is placing a higher value on the company's assets than what it would cost to replace them [98]. This can be an indication that the company has a strong brand, intellectual property, or other intangible assets that are difficult to replicate. In this study, Tobin's Q was calculated using a financial statement from the open-access DART database (https://dart.fss.or.kr, accessed on 1 November 2022).

Digital Marketing Innovation *(DI)*: We used customer engagement rates on social media as a proxy for digital marketing innovation because a high customer engagement rate on social media is a key metric in digital marketing innovation [99]. Social media is defined as digital communication platforms that allow parties to connect with each other to share information [54]. A high engagement rate on social media is an indication that a brand is effectively engaging with its audience, building relationships, and generating interest in its products or services. This can lead to building stronger relationships with their customers, increasing brand awareness, and, ultimately, sales.

Customer engagement rate (*ER*) is a metric that measures the level of engagement that a piece of content or a social media account has with its audience. Engagement rate is an

important metric because it provides an indication of how well a firm is connecting with its audience on social media. Drawing upon Culnan et al. [100] and Benitez et al. [101], we measured *ER* as a multidimensional construct of Facebook (a group of experts believes that for B2C companies, advertising is more effective on Facebook [102]) and YouTube features [103] using the following formula [104]:

$$ER = \frac{(Total\ Engagement)}{(Total\ Followers)} \times 100 \qquad (1)$$

where total engagement is calculated as a mean of various interaction metrics such as the number of likes, comments, shares, saves, and clicks on a particular post on Facebook and the sum of likes, dislikes, and comments on YouTube [35,105].

Marketing Capability (*MC*): Following prior literature [106,107], we used input-output stochastic frontier analysis (MC-IO SFA) to evaluate firm marketing capability. Stochastic frontier analysis (SFA) is a statistical method used to estimate the efficiency of a firm or organization. SFA measures the gap between the actual performance of a firm and the maximum possible performance given its resources and market conditions. This gap is known as the efficiency score and it can be used to identify areas where the firm can improve its performance. MC-IO SFA measures a company's marketing capability by comparing its actual marketing expenditures (inputs) such as total assets, selling, general, and administrative expenses and technological know-how reflected in patent stock to its actual sales revenue (outputs) while taking into account the external market environment and various external factors that may impact marketing performance. The analysis then generates a frontier that represents the maximum marketing efficiency that can be achieved by the company, given its inputs and the external factors. In this study, *MC* is calculated using input and output data from financial statements obtained from the open-access DART database (https://dart.fss.or.kr, accessed on 1 November 2022).

Firm Size (*FS*): Market capitalization is often used as a measure of company size and value because it reflects the market's perception of a company's future growth potential and profitability, which can be influenced by factors such as innovation, competitive advantage, and market share. While other measures such as total assets can also be useful indicators of a company's financial health and performance, market capitalization is often considered a more appropriate measure for assessing a company's growth potential and value creation, particularly for companies that are focused on innovation and have a strong track record of introducing new products or services. Market capitalization was chosen as a firm size measure in this study because market capitalization is an important measure that can provide insights into a company's size, growth potential, and value creation, particularly in the context of innovation and other factors that can impact a company's future prospects. It is often used to classify companies into different categories, such as large-cap, mid-cap, or small-cap, based on their relative sizes and potentials [108]. Market capitalization represents the total value of a company's outstanding shares of stock and is calculated by multiplying the current market price of each share by the total number of outstanding shares. In this study, *FS* is a continuous variable of market capitalization obtained from the open-access DART database (https://dart.fss.or.kr, accessed on 1 November 2022).

Descriptive statistics in Table 1 below provide an overview of the data.

**Table 1.** Summary statistics (N = 200).

| Variable | Mean | S.D. |
|---|---|---|
| Digital Marketing Innovation | 0.337 | 0.666 |
| Firm Performance | 0.563 | 0.469 |
| Marketing Capability | 0.531 | 0.284 |
| Firm Size (million KRW) | 157,120 | 384,979 |

## 4. Data Analysis

### 4.1. Model Formulation

A linear moderated mediation analysis [95] was performed to explore the impact of *DI* on *FP* through *MC* by *FS*. Following Edwards and Lambert [109], four linear equations in (2) and (3) below were estimated using OLS regression. This allowed us to moderate the direct and indirect effects of *DI* on *FP* through *MC* by *FS*.

$$FP = i_1 + c_1 DI + \varepsilon_1 \tag{2}$$

$$MC = i_2 + d_1 DI + \varepsilon_2 \tag{3}$$

$$MC = i_{MC} + b_1 DI + b_2 FS + b_3 DI * FS + \varepsilon_{MC} \tag{4}$$

$$FP = i_{FP} + a_1 DI + a_2 MC + \varepsilon_{FP} \tag{5}$$

where *MC* is marketing capability, *DI* is digital innovation, *FS* is firm size, *FP* is firm performance, and $\varepsilon$ is an error term.

We examined the influence of the moderator, firm size on the mediation effect, and digital marketing innovation on firm performance through marketing capability. The results of the proposed model (2), (3), (4) and (5) are presented in Table 2.

The indirect effect of *DI* on *FP* through *MC* (that is, the moderated mediation effect, *IE*) is the product of the conditional effect of *DI* on *MC* from Equation (4) and the effect of *MC* on *FP* controlling for *DI* in Equation (5) [110] as follows:

$$IE = b_1 a_2 + b_3 a_2 * FS. \tag{6}$$

Before we ran the moderated mediation analysis, we tweaked the data set a little bit. Following the prior literature [111], continuous variables were first mean centered. The mediation package in *R* was used to fit a moderated mediation model [95,111].

### 4.2. Results of Proposed Model

The model 1 estimates in Table 2 show that firms with higher marketing innovation tend to show higher firm performance (0.135, $p < 0.01$), in support of H1. The model 2 estimates in Table 2 indicate that firms with higher digital marketing innovation tend to have higher marketing capability (0.244, $p < 0.05$), in support of H2. The model 3 estimates in Table 2 show the existence of the mediation effect, that is, an indirect effect that the marketing capability affects the firm performance while controlling for the digital marketing innovation. In other words, firms with higher marketing capability tend to show higher firm performance (0.396, $p < 0.01$), in support of H3. These mediating effects are partial because direct effects of digital marketing innovations still exist after controlling for indirect effects.

**Table 2.** Results of Equations (2) and (3).

| Variables | Model 1 (DV = FP) Coeff. | Model 2 (DV = MC) Coeff. | Model 3 (DV = MC) Coeff. | Model 4 (DV = FP) Coeff. |
|---|---|---|---|---|
| DI | 0.135 ** (0.045) | 0.244 * (0.078) | 0.588 *** (0.134) | 0.894 ** (0.253) |
| FS | | | −0.626 (0.426) | |
| DI * FS | | | −0.395 *** (0.106) | |
| MC | | | | 0.396 ** (0.213) |
| Constant | 0.652 (0.017) | 0.222 (0.039) | −0.047 (0.041) | 0.243 ** (0.212) |
| | *Adj.* $R^2$ = 0.109 | *Adj.* $R^2$ = 0.309 | *Adj.* $R^2$ = 0.224 | *Adj.* $R^2$ = 0.309 |
| | *F*-statistic = 8.639, $p < 0.01$ | *F*-statistic = 5.794, $p < 0.05$ | *F*-statistic = 6.974, $p < 0.001$ | *F*-statistic = 9.853, $p < 0.001$ |

\* $p < 0.05$, ** $p < 0.01$, *** $p < 0.001$. Standard errors in parentheses.

A test of the moderator (firm size) on the mediation effect and digital marketing innovation on firm performance through marketing capability using Equation (6) yielded an indirect effect of firm size on the mediation effect as follows:

$$\text{Indirect effect} = 0.233 - 0.156 * FS,$$

which is a linear function of firm size with intercept 0.233 and slope $-0.156$.

Figure 2 shows these features graphically. The figure suggests that the indirect effect of digital marketing innovation on firm performance through marketing capabilities decreases as company size increases as the slope of the line—the index of moderated mediation—is negative. This could mean that smaller companies benefit more from digital marketing innovation and marketing capabilities in terms of improving their performance than larger companies do. Alternatively, it could indicate that larger companies have more diverse marketing capabilities, and digital marketing innovation may not be as significant a factor in their overall marketing strategy.

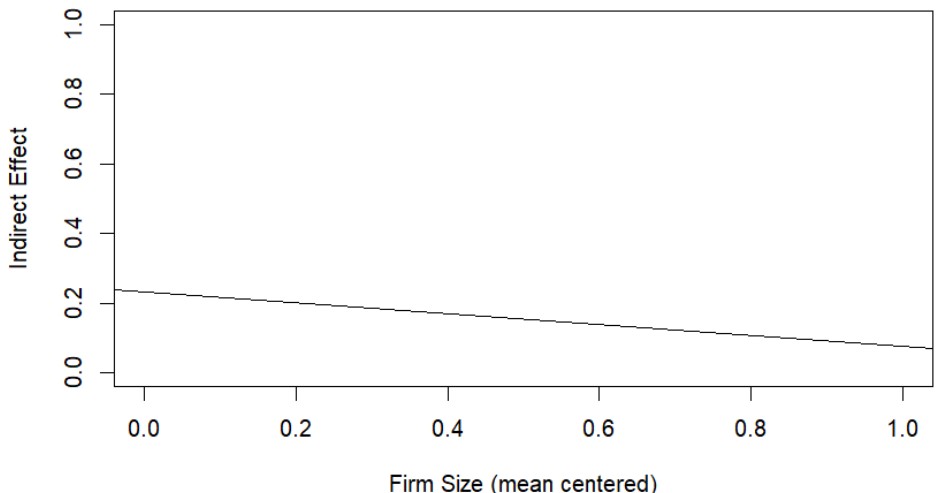

**Figure 2.** Indirect effect of firm size on the mediation effect.

The result that the 95% bootstrap confidence interval for the slope of this function does not include zero ($-0.331$ to $-0.011$) and has a negative upper bound indicates that the indirect effect of digital marketing innovation on firm performance through marketing capability is negatively moderated by firm size, in support of H5.

All the VIFs are less than 2.5, multicollinearity does not seem to be a problem in our analysis.

To understand the specific pathway through which digital marketing innovation affects firm performance through a moderated mediation effect, the average causal mediation effect (*ACME*) and average direct effect (*ADE*) were estimated with the quasi-Bayesian Monte Carlo method based on normal approximation [112]. To allow the mediation effects to be moderated by firm size, we set the data into large and small firms using KOSDAQ and KOSPI as dummies.

*ACME* refers to the average effect of *FS* on *FP* that is mediated through *MC*. *ACME* represents the effect of the *DI* on *FP* that operates through *MC*, while the direct effect represents the effect that operates independently of *MC*. On the other hand, the *ADE* represents the average effect of *DI* on *FP* that is not mediated through *MC*. Total Effect is the sum of the direct and indirect effects, and Prop. Mediated indicates how much the indirect effect explains the overall effects.

Results in Table 3 are consistent with ones in Table 2. There is a moderated mediation effect of firm size on the impact of digital marketing innovation on firm performance through marketing capability. Interestingly, ACME (0.304 ***) and ADE (0.111 ***) in Table 3 indicate that the indirect effect of digital marketing innovation on firm performance through marketing capability is greater than the direct effect. Our findings

imply that a company's ability to develop and use its marketing capability is critical to maximizing the impact of digital marketing innovations on company performance. Digital marketing managers should focus their investments and efforts on developing and improving their marketing capability by providing new tools, techniques, and channels to reach and engage with customers so that digital marketing innovation can more effectively increase firm performance.

**Table 3.** Results of causal mediation analysis.

| | Estimate |
|---|---|
| ACME | 0.304 *** |
| ADE | 0.111 *** |
| Total Effect | 0.415 *** |
| Prop. Mediated | 0.733 *** |

*** $p < 0.001$.

We assessed the robustness of our study by testing whether the results of our study were consistent when using different measures for firm size and performance. Total assets and revenue were used, respectively, for the measures of firm size and performance. The moderated mediation analysis with different measures confirmed the results consistent with previous analysis, as shown in Table 4.

**Table 4.** Results of robustness checks.

| | *(DV = MC)* | *(DV = Revenue)* |
|---|---|---|
| **Variables** | **Coeff.** | **Coeff.** |
| *DI* | 0.299 *** (0.061) | 0.159 ** (0.054) |
| Total Asset | −0.106 (0.033) | |
| DI * Total Asset | −0.587 *** (0.159) | |
| MC | | 0.120 + (0.065) |
| Constant | −0.079 (0.018) | 0.505 * (0.203) |
| | *Adj. $R^2$* = 0.257 | *Adj. $R^2$* = 0.109 |
| | *F*-statistic = 8.175, $p < 0.001$ | *F*-statistic = 4.777, $p < 0.05$ |

+ $p < 0.10$, * $p < 0.05$, ** $p < 0.01$, *** $p < 0.001$. Standard errors in Parentheses.

## 5. Discussions

With the rapid evolution of technology and the Internet, the way people consume information and make purchasing decisions is constantly changing. To stay relevant and competitive, businesses need to adapt their marketing strategies to these changes [113]. Digital marketing innovation is crucial for businesses that want to stay ahead of the competition and effectively reach their target audience in today's rapidly changing digital landscape [114]. While there has been a considerable amount of research on digital marketing, there is still relatively little empirical research specifically focused on the impact of digital marketing innovation and its conditions. Ref. [115] showed that firms that engage in digital marketing innovation have higher levels of performance, including increased sales growth and profitability. The study also found that factors such as organizational culture and managerial support play a role in the successful implementation of digital marketing innovation. Ref. [116] suggested that digital marketing innovation has a positive effect on customer engagement, and that the impact is even stronger when innovation is combined with personalization and interactivity. These studies provide some insights into the impact of digital marketing innovation, but more research is needed to fully understand the conditions under which digital marketing innovation can be most effective. As the digital landscape continues to evolve, there will be a growing need for research on this topic to help businesses stay competitive, effectively reach their target audiences, and enhance their performance.

Our findings make several contributions to the field of digital marketing innovation. There have been many prior studies on technology and product innovations. Because the concept of digital marketing innovation is modern and new, there are few studies that clearly show the impact of digital marketing innovations on company performance. Therefore, our study fills this gap by expanding the literature on digital marketing innovation.

Most of the articles in the previous literature prove the direct impact of innovation on the effectiveness of a company [40,41,117], but only a few researchers use marketing capabilities as a mediating effect between innovation and firm performance and use firm size as a moderator on the mediation effect. Therefore, we fill the research gap by suggesting that marketing capabilities are considered an important factor and play a decisive role in the linkage between marketing innovations and firm results, which is moderated by firm size.

In the innovation literature, SMEs and large companies were analyzed separately in most cases [118–120], but this study compares SMEs and large firms to show the moderating effect of firm size on the linkage between innovations and marketing capabilities.

## 6. Conclusions

This research examined the existence and process of the indirect effect of digital marketing innovation on firm performance through marketing capability moderated by firm size using 100 Korean firm samples. Our findings show that digital marketing innovation has a positive impact on firm performance through the improvement and growth of marketing capability, and this mediation effect is moderated by firm size. Further, the indirect effect of digital marketing innovation on firm performance through marketing capability is greater than the direct effect.

The results of this study have some managerial implications. Firstly, marketing managers should focus their efforts on the development of marketing capability to maximize the impact of digital marketing innovation on firm performance. Companies can leverage digital marketing tools and technologies to improve marketing effectiveness, reach and engage with new customers, and ultimately increase growth and profitability. Here are some of the ways that digital marketing innovation can impact firm performance through marketing capability. Firstly, digital marketing enables businesses to target and segment their marketing efforts more effectively, using data and analytics to identify and reach the right customers with the right messages. This can lead to increased conversion rates, higher customer engagement, and improved ROI. Secondly, digital marketing provides businesses with a range of tools and platforms to engage with customers in new and innovative ways, such as social media, email, and mobile marketing. By creating engaging and interactive experiences for customers, businesses can build stronger relationships and loyalty, which can lead to increased sales and revenue. Thirdly, our findings imply that the impact of digital marketing innovation on firm performance through marketing capability can be particularly relevant for Small and Medium-sized Enterprises (SMEs). SMEs often have limited resources and face greater challenges in adopting and implementing digital marketing strategies compared to larger firms. However, digital marketing innovation can also offer SMEs new opportunities to improve their marketing capabilities and enhance their competitiveness. By investing in digital marketing innovation, SMEs can improve their marketing capabilities and better reach and engage with customers through various digital channels. This can lead to increased customer satisfaction, loyalty, and sales, ultimately improving firm performance. However, the impact of digital marketing innovation on firm performance through marketing capability may depend on various factors, such as the nature of the industry, the level of competition, the resources available, and the level of expertise and knowledge of the firm's personnel. Therefore, SMEs need to carefully assess their digital marketing strategies and capabilities, and continuously monitor and adapt to changes in the market environment. By leveraging the opportunities provided by digital marketing, SMEs can compete more effectively in the market and achieve sustainable growth and success.

Here are some limitations of this study and suggestions for future research directions. First, in the prior literature, Karabulut [121] found that innovation has a negative impact on company growth. He also suggested that innovation would not be useful if neither the market, nor manufacturers, nor suppliers, nor consumers want to change the existing situation. Laforet [122] noted that uncontrolled business growth is ensured if the company does not consider the negative consequences of innovation. Future research should examine what exactly the negative effects of innovation are and what factors can help reduce these negative effects.

Second, this study uses engagement rates to measure digital marketing innovation, but a more accurate measurement of digital marketing innovation requires a multi-faceted approach incorporating a range of metrics and data points such as website traffic, social media engagement, search engine rankings, conversion rates, and innovation adoption rates. Measuring digital marketing innovation can be challenging, as it involves tracking and analyzing a range of metrics and data points across multiple channels and platforms. Therefore, this line of research appears to be a very promising avenue for future research.

Third, the relationship between innovation and firm performance may be affected by other factors, such as industry dynamics, market structure, and the nature of the innovation itself [123–125]. Therefore, it is important to consider these contextual factors when examining the relationship between innovation and firm performance, particularly when exploring the moderating role of firm size.

**Author Contributions:** S.-U.J. was responsible for conceptualization, model analysis, and writing. V.S. was responsible for data collection, literature review, and writing. All authors have read and agreed to the published version of the manuscript.

**Funding:** This research was supported by Hankuk University of Foreign Studies research fund.

**Institutional Review Board Statement:** Not applicable.

**Informed Consent Statement:** Not applicable.

**Data Availability Statement:** The detailed data used to support the findings of this study are available from the corresponding author upon request.

**Conflicts of Interest:** The authors declare no conflict of interest. The funders had no role in the design of the study; in the collection, analyses, or interpretation of data; in the writing of the manuscript; or in the decision to publish the results.

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
