# Peer review of "The Impact of Digital Marketing Innovation on Firm Performance: Mediation by Marketing Capability and Moderation by Firm Size"

_sustainability, doi:10.3390/su15075711_

Round 1
Reviewer 1 Report
Manuscript sustainability-2291800-peer-review:
The paper investigates the impact of digital marketing innovation on firm performance. It examines the mediating role of marketing capability and moderation by firm size. The topic is interesting and contemporary. The analysis is clear.
Comments:
|
Section |
Comment |
|
Introduction |
Reference needed: “the relationship between digital marketing innovation and firm performance is complex and interrelated with other factors such as firm size and marketing capability.” |
|
|
More discussion on the reasoning behind selecting the Korean context is needed |
|
|
The connection between the study contribution and sustainability should be more present in the paper. There is no mention of how this paper is related to Sustainability. |
|
Literature review |
“Hypothesis 2: Digital marketing innovation has a positive impact on marketing capability” Though explained in detail in the text, I wonder if Digital marketing innovation impacts marketing capability or the other way around. This will affect the results, you either need to test the reverse effect or provide evidence from previous literature about the actual direction of the relationship. |
|
|
The Literature review does not include any information about the Korean context, it is very important to build the whole paper to be coherent and interrelated to the proposed title. |
|
Materials and Methods |
“Table 1” Move after variable definition and computation, or explain how each variable is measured in the same section not afterward. |
|
|
“Figure 1” Move to the Hypothesis development section.
|
|
|
“Figure 2” the relationships in the figure do not reflect the hypothesis mentioned beforehand. Especially the presentation of the moderating and mediating effects. |
|
|
“Firm Performance (FP)” Mention the other measures of firm performance, even if you are using only one method. |
|
|
“Digital Marketing Innovation (DI): We use customer engagement rate on social media as a proxy for digital marketing innovation because a high customer engagement rate on social media is a key metric in digital marketing innovation.” Comment: Add references to support this measure or more deepen argument needs to be formulated to apply a new measure. |
|
|
“We measure ER as a multidimensional construct of Facebook and YouTube features” Comment: Are Facebook and YouTube the main platforms in Korea? Can this be proven? |
|
|
“Firm Size (FS): Market capitalization is a widely used measure of a company's value and financial performance, as it reflects both its current market price and the perceived future growth prospects.” Comment: Discuss the most common measure of size (Total Assets) and why it is not used. This can also be used as a robustness test. |
|
|
For the models of the study, I want to see the moderating effect in the Model Equation. Usually, this is added as a multiplication of variables. It is not clear how the paper treated the moderating variable in the OLS regression |
|
|
More caution should be provided when testing a mediation effect, either as partial or complete. This is not present in the analysis and the models provided. |
|
Conclusion |
More robustness checks should be implemented, especially for variables that have more than one measure (i.e. Size and Performance). The previous comments might impact the current results and hence the current conclusions. |
|
General Comments |
The paper is well written. However, I can’t see how the paper is is related to sustainability from the text, though this can be proven. I also did not see clear reasons why Korea is being chosen as a theory not data. Analysis needs improvements in the current measures and more robustness checks should be included. |
Author Response
We greatly appreciate the revision opportunity that you and the review team gave us. In order to address the review team’s concerns, we have made an earnest effort to revise our manuscript in both conceptual and empirical aspects. We are most grateful for the clear, thorough, specific, and constructive comments and suggestions that we received during the review process. Your invaluable guidance and support have helped us tremendously in the revision. Hope our revision meets with your requirement for publication in Sustainability.
REVISION NOTES
Paper Title: “The impact of digital marketing innovation on firm performance: Mediation by marketing capability and moderation by firm size”
Manuscript ID: 2291800
Responses to Reviewer #1’s Comments and Suggestions:
Reference needed: “the relationship between digital marketing innovation and firm performance is complex and interrelated with other factors such as firm s07ize and marketing capability.”
Thank you for your comment. The following references are incorporated.
Grewal, D.; Hulland, J.; Kopalle, P.K.; Karahanna, E. The future of technology and marketing: A multidisciplinary perspective. Journal of the Academy of Marketing Science 2020, 48, 1-8.
Sinha, K.K.; Saunders, C.; Raby, S.; Dewald, J. The moderating role of previous venture experience on breadth of learning and innovation and the impacts on SME performance. International Journal of Entrepreneurial Behavior & Research 2022, 28(2), 346-367.
More discussion on the reasoning behind selecting the Korean context is needed.
Your point is well taken. On page 2, we added a discussion of why Korean context was chosen in our study.
The connection between the study contribution and sustainability should be more present in the paper. There is no mention of how this paper is related to Sustainability.
Thank you for your insightful comment. We added to the Introduction section an explanation of why our research work on the impact of digital marketing innovation is closely linked to company sustainability.
“Hypothesis 2: Digital marketing innovation has a positive impact on marketing capability”
Though explained in detail in the text, I wonder if Digital marketing innovation impacts marketing capability or the other way around. This will affect the results, you either need to test the reverse effect or provide evidence from previous literature about the actual direction of the relationship.
Thank you for your insightful comment. We agree with a potential endogeneity issue you pointed out. On page 3 and 7, We added an explanation about the direction of the relationship.
The Literature review does not include any information about the Korean context, it is very important to build the whole paper to be coherent and interrelated to the proposed title.
Your point is well taken. We added information about the Korean context to the Literature Review section.
“Table 1” Move after variable definition and computation, or explain how each variable is measured in the same section not afterward.
Thank you for your suggestion. Table 1 is moved to the variable measurement section.
“Figure 1” Move to the Hypothesis development section.
Thank you for your suggestion. Figure 1 is moved to the Hypothesis development section.
“Figure 2” the relationships in the figure do not reflect the hypothesis mentioned beforehand. Especially the presentation of the moderating and mediating effects.
Thank you for your comment. As we think that the hypothesis is sufficiently explained in the conceptual model of Figure 1, Figure 2 is removed in the manuscript.
“Firm Performance (FP)” Mention the other measures of firm performance, even if you are using only one method.
Thank you for your comment. On page 7, other measures of firm performance are mentioned.
“Digital Marketing Innovation (DI): We use customer engagement rate on social media as a proxy for digital marketing innovation because a high customer engagement rate on social media is a key metric in digital marketing innovation.”
Comment: Add references to support this measure or more deepen argument needs to be formulated to apply a new measure.
Thank you for your comment. We added the following reference.
Kumar, A.; Kesharwani, S.; Mishra, A. Social media metrics and business performance: A meta-analysis. Journal of Business Research 2017, 70, 118-126.
“We measure ER as a multidimensional construct of Facebook and YouTube features”
Comment: Are Facebook and YouTube the main platforms in Korea? Can this be proven?
Thank you for your comment. Facebook and Youtube are one of the most popular social media platforms in Korea. According to data from Statista, Facebook had approximately 18.4 million monthly active users in South Korea as of January 2021. YouTube is also a popular platform in Korea, with a large number of content creators and viewers. According to a report by eMarketer, YouTube was the second most popular social media platform in Korea in 2020, with approximately 29.9 million users. This represents a significant increase from the previous year, indicating that the platform's popularity in the country is growing.
“Firm Size (FS): Market capitalization is a widely used measure of a company's value and financial performance, as it reflects both its current market price and the perceived future growth prospects.”
Comment: Discuss the most common measure of size (Total Assets) and why it is not used. This can also be used as a robustness test.
Thank you for your comment. Your point is well taken. Discussion about why market capitalization is a more relevant measure of firm size in our study than other measures such as Total Assets is added to our study on page 8. On page 12, robustness test is conducted with Total assets as another firm size measure.
For the models of the study, I want to see the moderating effect in the Model Equation. Usually, this is added as a multiplication of variables. It is not clear how the paper treated the moderating variable in the OLS regression. More caution should be provided when testing a mediation effect, either as partial or complete. This is not present in the analysis and the models provided.
Thank you for your comment. Your point is well taken. To make it clear, we included all detailed statistical procedures, results and explanations to lead to the moderated mediation model on page 8-9. Moderating effect is addressed as a multiplication of DI and FS variables in equation (4).
More robustness checks should be implemented, especially for variables that have more than one measure (i.e. Size and Performance). The previous comments might impact the current results and hence the current conclusions.
We agree that robustness checks are necessary. Following your suggestion, robustness checks are conducted and the results are added as Table 3 on page 11.
The paper is well written. However, I can’t see how the paper is related to sustainability from the text, though this can be proven. I also did not see clear reasons why Korea is being chosen as a theory not data.
Thank you for your comment. On page 2, we added the explanations.
We are most grateful for your comments and the chance to revise our work. With the suggested improvement of our paper, we hope you like the changes we have made in this revision and look forward to receiving your further comments. Hope this revision meets with the requirement for publication in Sustainability.

Reviewer 2 Report
Dear Authors,
Your paper "The impact of digital marketing innovation on firm performance: Mediation by marketing capability and moderation by firm size" sent to Sustainability Journal needs the following improvements.
Please try to rewrite some paragraphs from your manuscript because, during my initial documentation for this review, I found some similarities.
For example, at page 10, the paragraphs starting with "This function is depicted graphically in Figure 3. As can..." and "A 95% bootstrap confidence interval for this index—the slope..." are very similar to the article available at the address https://doi.org/10.1080/00273171.2014.962683.
The Introduction is almost complete. You should only include the clear description of the research gap and the research questions, because the readers are very interested in these aspects from the very beginning of the article.
In the last paragraph of the page 2, you say: "Today there are two types of marketing: traditional and digital marketing [15,15]." Please avoid using the same reference (in this case [15]) twice in the same sequence.
I recommend you to move the Figure 1 at the end of the Literature Review chapter, immediately after the development of the hypotheses.
I also recommend you to include in your work the following resources: https://doi.org/10.3390/admsci11030071 (size of business and digital marketing), https://doi.org/10.15611/aoe.2022.1.11 (peer communication), https://doi.org/10.3390/fi11060130 (marketing actions), https://doi.org/10.3390/admsci8030031 (firms' size). These references will improve the general context of your research.
At page 6 you say: "Data is collected from KRX data information system (https://data.krx.co.kr), which provides useful statistical information in various fields, from listed stocks to indices, securities products, derivatives and general products. Descriptive statistics in Table 1 below provide an overview of the data." For me, it is a bit unclear how you obtained the values of "Marketing Capability". Can you shortly explain the variables? Does KRX have a variable named "Marketing Capability" for every company?!?
At page 9, the numbers of the equations (2) and (3) should be correctly aligned with the equations. Please revise this editing issue.
For the regression analysis, please provide some information about the multicollinearity test.
Before the "5. Conclusions" chapter, I recommend you to include a "Discussions" chapter and here you should present your own results by comparing them to the others from the scientific literature. This way, you will highlight your own contribution to the field of knowledge.
Best Regards!
Author Response
We greatly appreciate the revision opportunity that you and the review team gave us. In order to address the review team’s concerns, we have made an earnest effort to revise our manuscript in both conceptual and empirical aspects. We are most grateful for the clear, thorough, specific, and constructive comments and suggestions that we received during the review process. Your invaluable guidance and support have helped us tremendously in the revision. Hope our revision meets with your requirement for publication in Sustainability.
REVISION NOTES
Paper Title: “The impact of digital marketing innovation on firm performance: Mediation by marketing capability and moderation by firm size”
Manuscript ID: 2291800
Responses to Reviewer #2’s Comments and Suggestions:
Your paper "The impact of digital marketing innovation on firm performance: Mediation by marketing capability and moderation by firm size" sent to Sustainability Journal needs the following improvements.
Please try to rewrite some paragraphs from your manuscript because, during my initial documentation for this review, I found some similarities. For example, at page 10, the paragraphs starting with "This function is depicted graphically in Figure 3. As can..." and "A 95% bootstrap confidence interval for this index—the slope..." are very similar to the article available at the address https://doi.org/10.1080/00273171.2014.962683.
Thanks for your comments. We revised and rewrote it accordingly.
The Introduction is almost complete. You should only include the clear description of the research gap and the research questions, because the readers are very interested in these aspects from the very beginning of the article.
Thank you for your comment. The research gap and the research questions are incorporated on page 2.
In the last paragraph of the page 2, you say: "Today there are two types of marketing: traditional and digital marketing [15,15]." Please avoid using the same reference (in this case [15]) twice in the same sequence.
Thank you for pointing out this mistake. We have corrected the references.
I recommend you to move the Figure 1 at the end of the Literature Review chapter, immediately after the development of the hypotheses.
Thank you for your suggestion. Figure 1 is moved to the end of the Literature Review section.
I also recommend you to include in your work the following resources: https://doi.org/10.3390/admsci11030071 (size of business and digital marketing), https://doi.org/10.15611/aoe.2022.1.11 (peer communication), https://doi.org/10.3390/fi11060130 (marketing actions), https://doi.org/10.3390/admsci8030031 (firms' size). These references will improve the general context of your research.
Thank you for your suggestion.To improve the general context of our research, we used the following information from the research papers listed above. https://doi.org/10.3390/admsci11030071: “But, Pollák and Markovič conclude in their research that most organizations are not ready to invest in marketing communications based on digital marketing, because it is still perceived by many firms as a supplement to traditional marketing.” https://doi.org/10.3390/fi11060130: “A group of experts believes that for B2C companies, advertising is more effective on Facebook.” https://doi.org/10.3390/admsci8030031: “The results of an empirical study by Joensuu-Salo et al. prove that marketing capabilities have a direct impact on firm performance.”
At page 6 you say: "Data is collected from KRX data information system (https://data.krx.co.kr), which provides useful statistical information in various fields, from listed stocks to indices, securities products, derivatives and general products. Descriptive statistics in Table 1 below provide an overview of the data." For me, it is a bit unclear how you obtained the values of "Marketing Capability". Can you shortly explain the variables? Does KRX have a variable named "Marketing Capability" for every company?!?
Thank you for your comment. More explanations about the operationalization of “Marketing Capability” is added on page 7. We use the input-output stochastic frontier analysis (MC-IO SFA) to evaluate firm marketing capability. Stochastic frontier analysis (SFA) is a statistical method used to estimate the efficiency of a firm or organization. MC-IO SFA measures a company's marketing capability by comparing its actual marketing expenditures (inputs) such as total assets, selling, general, and administrative expenses and technological know-how reflected in patent stock to its actual sales revenue (outputs), while taking into account the external market environment and various external factors that may impact marketing performance. The analysis then generates a frontier that represents the maximum marketing efficiency that can be achieved by the company, given its inputs and the external factors. In this study, MC is calculated using input and output data from financial statements obtained from the open access DART database.
At page 9, the numbers of the equations (2) and (3) should be correctly aligned with the equations. Please revise this editing issue.
Thank you for your comment. We revised it accordingly.
For the regression analysis, please provide some information about the multicollinearity test.
Thank you for your comment. On page 10, VIF is reported for the multicollinearity test.
Before the "5. Conclusions" chapter, I recommend you to include a "Discussions" chapter and here you should present your own results by comparing them to the others from the scientific literature. This way, you will highlight your own contribution to the field of knowledge.
Thank you for your insightful comments. “5. Conclusions” chapter is now incorporated into the manuscript before “6. Discussion” chapter.
We are most grateful for your comments and the chance to revise our work. With the suggested improvement of our paper, we hope you like the changes we have made in this revision and look forward to receiving your further comments. Hope this revision meets with the requirement for publication in Sustainability.

Round 2
Reviewer 1 Report
Comments are taken into account and the revised version is satisfactory in its current form.
Author Response
We are most grateful for your comments and the chance to revise our work.
Hope this revision meets with the requirement for publication in Sustainability.
Reviewer 2 Report
Dear Authors,
I appreciate your effort to improve the manuscript's quality. At this moment, I have the following remarks:
- at page 9, in the section "4.1. Model formulation", the equations (3) and (4) are wrapped within the text. Please correct this issue.
- at page 14, the first row is "several contributions to the field of digital marketing innovation". It is not a sentence and it starts with lower-case. Please revise this row.
- the Literature Review section should be improved by including the following resources: https://doi.org/10.3390/app9183685, https://doi.org/10.1016/j.indmarman.2020.07.022, https://doi.org/10.15611/aoe.2022.1.11, https://doi.org/10.1016/j.ibusrev.2021.101946.
- at page 9, there is the following sentence: "Indirect effect of DI on FP through MC (that is, moderated mediation effect, IE) is the product of the conditional effect of DI on MC from equation (1) and the effect of MC on FP controlling for DI in equation (2) [107] as follows:". But the equation 1 (eq 1) is not presented in the text. Please add the corresponding description.
Best regards!
Author Response
We greatly appreciate the revision opportunity that you and the review team gave us. In order to address the review team’s concerns, we have made an earnest effort to revise our manuscript in both conceptual and empirical aspects. We are most grateful for the clear, thorough, specific, and constructive comments and suggestions that we received during the review process. Your invaluable guidance and support have helped us tremendously in the revision. Hope our revision meets with your requirement for publication in Sustainability.
REVISION NOTES
Paper Title: “The impact of digital marketing innovation on firm performance: Mediation by marketing capability and moderation by firm size”
Manuscript ID: 2291800
Responses to Reviewer #2’s Comments and Suggestions:
- at page 9, in the section "4.1. Model formulation", the equations (3) and (4) are wrapped within the text. Please correct this issue.
Thank you for your comment. We revised it accordingly.
- at page 14, the first row is "several contributions to the field of digital marketing innovation". It is not a sentence and it starts with lower-case. Please revise this row.
Thank you for your comment. We revised it accordingly.
- the Literature Review section should be improved by including the following resources: https://doi.org/10.3390/app9183685, https://doi.org/10.1016/j.indmarman.2020.07.022, https://doi.org/10.15611/aoe.2022.1.11, https://doi.org/10.1016/j.ibusrev.2021.101946.
Thank you for your suggestion.To improve the Literature Review section of our research, we used the following information from the research papers listed above. https://doi.org/10.15611/aoe.2022.1.11: Today, social media is a key element of companies' marketing strategy. There is a strong correlation between customers' use of social media and their brand loyalty: a high level of usage yields higher loyalty. Moreover, a higher level of loyalty leads to a better sales performance. https://doi.org/10.3390/app9183685: There are different types of innovation. In our study, we will look at marketing innovation. Marketing innovation is the introduction of a new method of promoting, evaluating or selling goods/services, significant changes in the aesthetic design or packaging of goods. https://doi.org/10.1016/j.indmarman.2020.07.022: The mere possession of digital resources is not sufficient to create value and competitive advantage, so the focus should be on how to use digital resources as inputs, which in turn create capabilities. Social media is defined as digital communication platforms that allow parties to connect with each other, to share information.
- at page 9, there is the following sentence: "Indirect effect of DI on FP through MC (that is, moderated mediation effect, IE) is the product of the conditional effect of DI on MC from equation (1) and the effect of MC on FP controlling for DI in equation (2) [107] as follows:". But the equation 1 (eq 1) is not presented in the text. Please add the corresponding description.
Thank you for your comment. We revised it accordingly.
We are most grateful for your comments and the chance to revise our work. With the suggested improvement of our paper, we hope you like the changes we have made in this revision and look forward to receiving your further comments. Hope this revision meets with the requirement for publication in Sustainability.